# Novel Characteristics of Race-Specific Genetic Functions in Korean CADASIL

**DOI:** 10.3390/medicina55090521

**Published:** 2019-08-22

**Authors:** Yerim Kim, Seung-Hoon Lee

**Affiliations:** 1Department of Neurology, Kangdong Sacred Heart Hospital, College of Medicine, Hallym University, Seoul 05355, Korea; 2The Korean Cerebrovascular Research Institute, Seoul 03100, RKorea; 3Department of Neurology, Seoul National University Hospital, Seoul 03080, Korea

**Keywords:** CADASIL, mutation, NOTCH 3 protein, stroke, cerebral infarction, intracranial hemorrhage

## Abstract

*Background and Objectives:* Previous studies found differences in the characteristics of NOTCH3 mutations in Caucasians and Asians with cerebral autosomal dominant arteriopathy with subcortical infarcts and leukoencephalopathy (CADASIL). Therefore, we sought to investigate the correlations between genetic and clinical/radiological findings in Korean CADASIL patients including some variants of unknown significance (VUS). *Materials and Methods:* We screened 198 patients with a suspected diagnosis of CADASIL between 2005 and 2015 via Sanger sequencing. *Results:* A total of 34 subjects (52.5 ± 9.5 years) were included. The majority of the mutations were in exon 3 and exon 11. R75P mutations (n = 5), followed by Y465C and R544C mutations (n = 4) were the most prevalent. Patients with those mutations exhibited less frequent anterior temporal (AT) or external capsular (EC) hyperintensities compared to patients with other locus mutations. Hemorrhagic stroke (HS) was found to be associated with mutations in exon 3 (R75P), exon 9 (Y465C), exon 11 (R587C), and exon 22 (R1175W variants), which were common locations in our study. Although it is unclear that genetic differences might affect the phenotypes in ethnicities, Asian population shows less migraine or seizure, but more intracerebral hemorrhage. Unlike in westernized countries, typical AT or EC hyperintensities may not be significant MRI markers, at least in Korean CADASIL patients. Furthermore, similar to R75P phenotypes, it is a novel finding that patients with Y465C and R1175W VUS have less frequent AT involvement than Caucasians. *Conclusion:* The associations between HS and common genetic locations account for the increased development of intracerebral hemorrhage in Koreans rather than Caucasians. We suggest that some CADASIL mutations appear to impart novel region-specific characteristics.

## 1. Introduction

Cerebral autosomal dominant arteriopathy with subcortical infarcts and leukoencephalopathy (CADASIL) is a rare hereditary disease caused by mutations in the NOTCH3 gene, which encodes for the first five epidermal growth factor repeats on chromosome 19p13.12 (Figure 1A,B). CADASIL is characterized by severe white matter changes (WMCs), especially in the anterior temporal (AT) and external capsule (EC) lesions. The main clinical symptoms are recurrent ischemic stroke (IS), migraine, and progressive dementia [1].

While the majority of NOTCH3 gene mutations were found in exon 4 in Caucasians [2,3] and in Chinese mainland [4], those were in exon 11 in Han Chinese in Taiwan [5] and in Koreans [6,7]. In contrast to westernized countries, an infrequent involvement of AT hyperintensities on T2-weighted magnetic resonance imaging (MRI) was also observed in Asian studies [5,8].

Although the underlying genetic mechanisms remain unclear, a specific gene has unique roles, and so changes in amino acids via gene mutations may provoke alterations that lead to region-specific functional deficits. Therefore, we sought to investigate the correlations between genetic and clinical/radiological findings in Korean CADASIL patients including some variants of unknown significance (VUS).

## 2. Materials and Methods

We screened 198 consecutive patients with a suspected diagnosis of CADASIL between 2005 and 2015 at Seoul National University Hospital (SNUH). Genetic work-ups were performed based on the physicians’ discretion considering the previous medical history, neurological examination, and MRI findings. The inclusion criteria were severe white matter hyperintensities on the brain MRI and at least one of the following: recurrent subcortical infarction at relatively young age, cognitive impairment, psychiatric symptoms, migraine, seizure, and family history of strokes. Patients with definite etiologies of large artery stenosis or cardioembolism for index stroke lesions were excluded. Among them, 34 patients were diagnosed as having known NOTCH3 mutations (n = 23) or novel VUS (n = 11).

Mutational hotspots of the NOTCH3 gene in exons 2–23 were screened via Sanger sequencing. Genomic deoxyribonucleic acid (DNA) was extracted from peripheral blood. The polymerase chain reaction sequences were compared with the previously published NOTCH3 DNA sequences. To identify the likely significance of known NOTCH3 DNA sequences or variants, we examined published literature, the “NOTCH3 gene homepage-Leiden Open Variation Database 3.0” (Leiden University Medical Center, Leiden, The Netherlands), and “Exome Aggregation Consortium database”.

1.5 Tesla or 3.0 Tesla MRI was performed in almost patients except two subjects. One was a daughter of a symptomatic patient. Since she did not have any clinical symptoms, she did not want to take brain image. Although another subject did not have MRI image, MRI results from other hospitals were available only on chart records. The WMCs and cerebral microbleeds (CMBs) were assessed with a scoring system by using modified Fazekas scales that was based on previous literatures [9]. We analyzed the AT and EC lesions, which are known to be useful markers for CADASIL. The institutional review board (Approval number: H-1504-064-664) approved this study protocol, and written informed consent was obtained from all participants.

## 3. Results

Among a total of 34 subjects, 11 known mutations (n = 23) and 8 VUS (n = 11) were identified. Ischemic stroke (IS) was the most common symptom (n = 15), and the other manifestations were HS (n = 6) and migraine (n = 8). Seizure was not detected in our study population. Hemorrhagic stroke (HS) was found to be associated with mutations in exon 3 (R75P), exon 9 (Y465C), exon 11 (R587C), and exon 22 (R1175W VUS) (see Table 1).

The majority of the mutations or VUS were in exon 3 and exon 11. R75P mutations (n = 5), followed by Y465C and R544C mutations (n = 4) were the most prevalent (Figure 1C and Appendix A). Although the enrolled patients were gathered from multiple areas, three of the four CADASIL patients with R544C mutation were born on Jeju Island.

Among the five subjects with R75P mutations or R75Q VUS, only one individual showed AT hyperintensities. In four patients with Y465C mutations, none had ischemic strokes. Among them, three patients are one family. Among the four subjects with R544C mutations, three patients did not have any strokes and did not show AT or EC hyperintensities (Table 1).

While all patients with mutations in exon 4 had AT lesions, approximately less than a third of patients with mutations in some mutational hotspots had AT lesions; R75P/R75Q VUS (16.7%), Y465C (0%), R544C (25%), and R1175W VUS (33.3%) (see Figure 2). As a result, compared to mutations in exon 4, the proportion of AT involvement in patients with the most prevalent loci was 17.6% (Figure 3A,B). Results with EC hyperintensities also showed similar distribution.

The genetic spectrum, clinical manifestations, and MRI findings of CADASIL in different articles were compared in Table 2.

## 4. Discussion

Unlike previous reports from western countries, our results demonstrated that the majority of mutations were in exon 11 followed by exon 3. The majority of mutations (R75P, Y465C, R544C, and R1175W VUS) were associated with less prevalent AT or EC hyperintensities. HS was found to be associated with R75P, Y465C, R587C mutations, and R1175W VUS, which were common loci in our study. These results suggest that the characteristics of NOTCH3 mutations in Koreans may be different from those in Caucasians.

According to previous Caucasian articles, NOTCH3 mutations were frequently found in exons 2–6, especially in exon 4 [1,2,10,11]. In British cases, 73% of NOTCH3 mutations were in exon 4 [2]. However, in 20 subjects in 17 Korean CADASIL families on Jeju Island, 85% of NOTCH3 mutations were in exon 11 and R544C mutations were present in 75% of patients [7]. This predominance of the R544C mutation is quite interesting. Three of four CADASIL patients with the R544C mutation in exon 11 were born on Jeju Island, which suggests a ‘founder effect’. This founder has been reported in small, isolated islands or inlands such as Taiwan (R544C) [5], Finland (R133C) [3], and mid-Italy (R607C) [12] (Figure 4). There are some Taiwan studies that analyze stroke etiology and radiological findings focused on R544C mutations [13,14]. According to a prospective, multicenter study in Taiwan, small-vessel occlusion was significantly associated with R544C mutation [13]. Another Taiwan study demonstrated that of 67 patients with R544C mutations, patients with intracerebral hemorrhage (ICH) onset were more likely to have white matter hyperintensities and higher total cerebral MBs [14]. Despite a lot of controversy, R75P could be considered the best explained cystein sparing Notch3 mutation to date [6,8]. Furthermore, recent study demonstrated another cystein sparing Notch3 mutation (D80G) in four German families [15]. These findings are evaluated as having broadened insight in CADASIL.

There were no patients that presented with seizures, which is in accordance with previous literatures [5,6]. While approximately half of patients reported migraines in British (54.2%) [2] and American (40%) groups, migraine appeared to be less prevalent in Asians (Chinese, 4.8%; Korean, 10%) [5]. Most of the patients with R75P mutations/R75Q VUS (IS, n = 4; HS, n = 1) demonstrated strokes as clinical symptoms. Although the exact patho-mechanisms of intracerebral hemorrhage (ICH) in CADASIL patients are not fully elucidated yet, MBs and antithrombotics seem to be associated with the increased risk of ICH. Notably, HS was found to be associated with mutations in exon 3 (R75P), exon 9 (Y465C), exon 11 (R587C), and exon 22 (R1175W VUS), which were common locations in this study. Compared to those patients, none of the patients with NOTCH3 mutations in exon 4 had HS. These findings may account for the increased development of ICH in Korean individuals rather than Caucasians [6]. In this study, although the association between R544C and HS has not been proven, it is thought that the number of enrolled patients with R544C is too small.

AT and EC hyperintensities have been regarded as significant markers for Caucasian CADASIL patients, with a sensitivity of 89% [2]. However, this was not consistent with the results of an Asian study (42.9% in Chinese and only 20% in Korean) [5,7]. There were typical AT hyperintensities in only one of six subjects with R75P mutations/R75Q VUS and one of four patients with R544C mutations. Previous articles demonstrated that R75P mutations were associated with an infrequent involvement of AT lesions, thus widening the spectrum of CADASIL [6,8]. It is noteworthy that these types of NOTCH3 mutations occurred in the majority of Korean CADASIL patients. Therefore, we suggest that AT or EC lesions may not be significant MRI markers, at least in Korean CADASIL.

In addition, it is a novel finding of this study that patients with Y465C and R1175W VUS have low AT involvement. However, the number of patients included is too small to generalize this result.

Conventionally, cerebral WMCs have been indicated in microangiopathy-related cerebral damage and have shared patho-mechanisms with CMBs. However, there were no definite linear correlations between the number of CMBs and the Fazekas scales (Spearman’s *r* = 0.51). These observations suggest that WMCs in CADASIL patients result from a different pathophysiology than conventional small-vessel diseases. Because the AT pole has a unique rotatory structure and is vascularized by anterior temporal artery, AT hyperintensities are enlarged perivascular spaces that exhibit myelin degeneration and subsequent axonal disruption, and are not lacunar infarctions [11,16].

There are some caveats. First, because this was a small single-center study, our findings may not be applicable to other populations. Second, we conducted NOTCH3 genetic sequencing in patients with a suspected diagnosis of CADASIL or their relatives. Some asymptomatic CADASIL carriers may have been excluded. Third, this was a retrospective study. Since most patients were examined during the acute stroke period, cognitive evaluations were not sufficient. Finally, we may have missed or overestimated the significance of mutations or VUS. However, to evaluate mutations accurately, we searched both published articles and NOTCH3 gene databases.

## 5. Conclusions

In conclusion, unlike in westernized countries, typical AT or EC hyperintensities may not be significant MRI markers, at least in Korean CADASIL patients. The associations between HS and common locations in our study may account for the increased development of ICH in Korean individuals rather than Caucasians. Although the underlying genetic mechanisms remain unclear, we suggest that some CADASIL mutations appear to impart novel region-specific characteristics. To clarify the variable functional roles of NOTCH3 mutations, more cell expression studies may be needed.

## Figures and Tables

**Figure 1 medicina-55-00521-f001:**
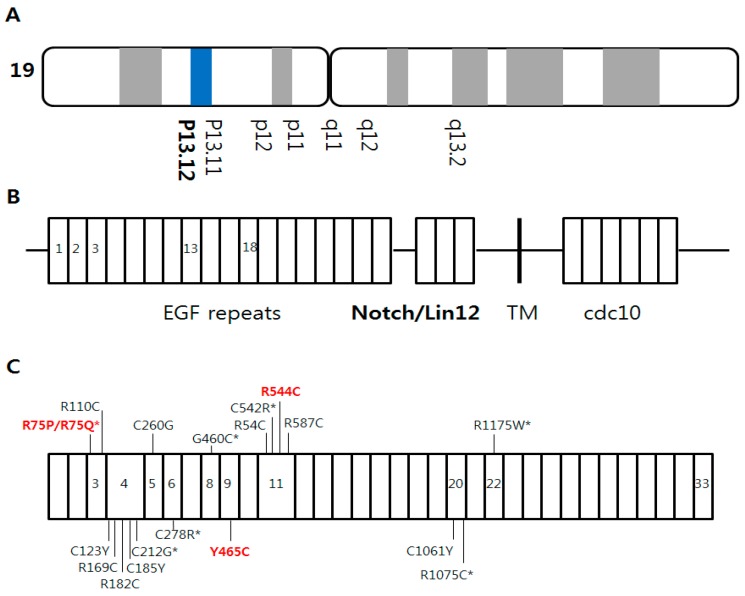
NOTCH3 gene mutations. (**A**) NOTCH3 gene located on chromosome 19p13.12; (**B**) NOTCH3 gene mutations are located within the extracellular domain that encodes the epidermal growth factor like repeats, 3 Lin-NOTCH repeats, and 1 transmembrane region; (**C**) NOTCH3 gene mutations or variants of unknown significances in our study. Red bold text is the most common genetic region in this study. * Novel variants of unknown significance.

**Figure 2 medicina-55-00521-f002:**
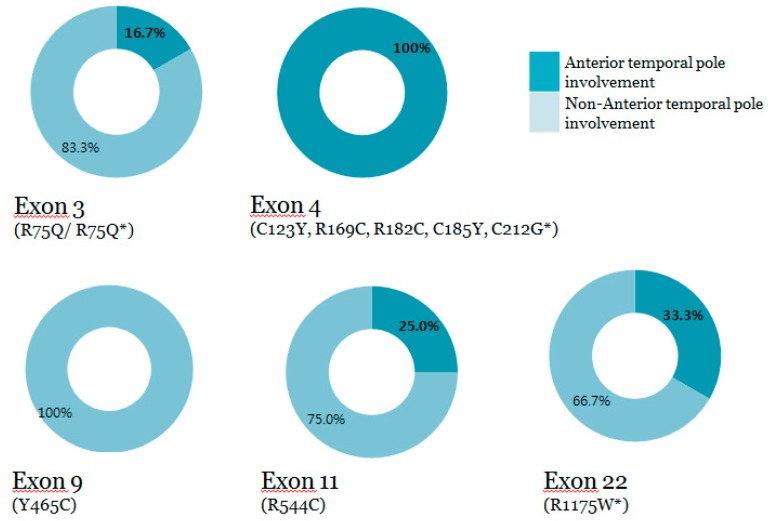
Anterior temporal pole involvement according to specific exon locations.

**Figure 3 medicina-55-00521-f003:**
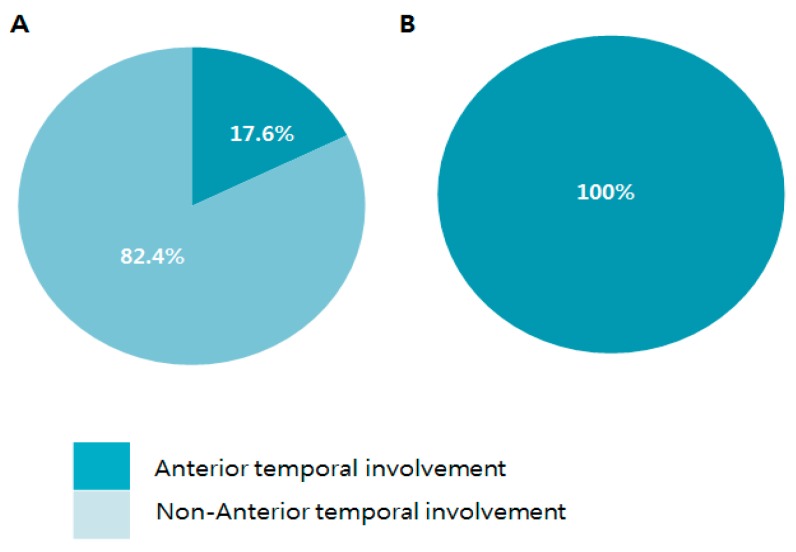
Anterior temporal pole involvement in prevalent NOTCH3 mutations compared to mutations in Exon 4. (**A**) NOTCH3 gene mutations of R75P, Y465C, R544C, and R1175W variants of unknown significance (VUS) and (**B**) NOTCH3 gene mutations in exon 4 (C123Y, R169C, R182C, and C185Y).

**Figure 4 medicina-55-00521-f004:**
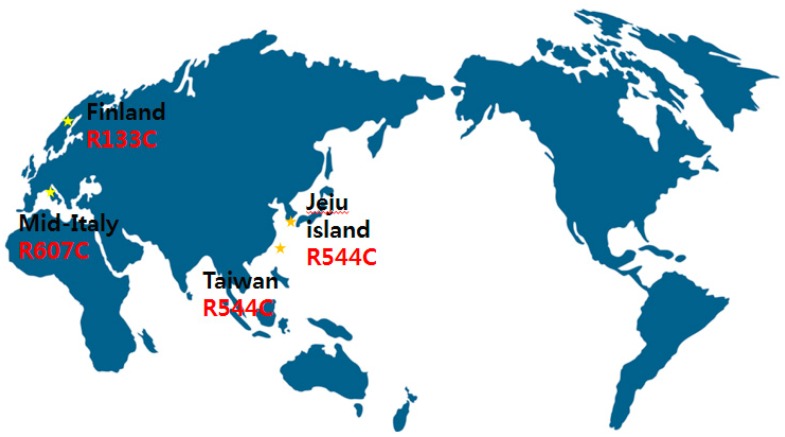
‘Founder effect’ reported in many countries.

**Table 1 medicina-55-00521-t001:** Baseline characteristics of patients with known NOTCH 3 mutations or novel variants of unknown significance (VUS).

	Age/Sex	Nucleotide	Exon	Amino Acid	Risk Factors	Symptom	MRI/MRA	Location of IS	Location of HS	No. of MBs	Location of MBs	Modified Fazekas	White Matter Lesions
1	57/M	224G>C	3	R75P	DL	IS	+/+	Rt. pons		12	B,D,E	3	a1,b1,c
2	43/M	224G>C	3	R75P	HTN, DL	IS	+/+	Rt. MCA		1	A	2	c
3	43/F	224G>C	3	R75P	HTN	HS	+/+		Rt. BG	0		1	c
4	56/F	224C>T	3	R75P		–	+/+			0		1	c
5	64/F	224C>T	3	R75P	DL	IS	+/+	Lt. CR		16	B,C,E	3	b1,c
6	57/F	224G>A	3	R75Q*	HTN, DM, DL	IS	+/+	Lt. thalamus		9	A,B,E	2	c
7	34/F	328C>T	3	R110C		–	+/+			0		2	a1,b1,c
8	44/M	368G>A	4	C123Y		IS	+/+	CR		14	C,D,E	3	a2, b1,c
9	59/M	505C>T	4	R169C		IS	+/+	Cerebellum		4	B	3	a2,b2,c
10	57/F	544C>T	4	R182C	HTN	IS	+/+	Lt. BG		3	B	3	a2,b2,c
11	62/F	554G>A	4	C185Y		migraine	+/+					3	a2,b2,c
12	62/F	634T>G	4	C212G*	HTN	IS, migraine	+/+	Lt. CR		0		2	a2, b1,c
13	49/M	778T>G	5	C260G	HTN, DL, SM	IS	+/+	Rt. CR		0		2	a2,c
14	60/F	832T>C	6	C278R*	DL	IS	+/+	pons		24	B,C,E	3	a2,b2,c
15	51/F	1378G>T	8	G460C*	HTN	migraine	+/+			15	A,B,C,D,E	2	a2, b1,c
16	44/F	1378G>T	8	G460C*	HTN, DL	migraine	+/+			0		2	b1,c
17	52/F	1394A>G	9	Y465C	HTN	–	+/+			2	B,E	3	b1,c
18	30/F	1394A>G	9	Y465C		migraine	−/−	N/A	N/A	N/A	N/A	N/A	N/A
19	52/F	1394A>G	9	Y465C	DM	–	+/+			0		3	c
20	46/M	1394A>G	9	Y465C		HS	+/+		Lt. thalamus	0		1	c
21	52/M	160C>T	11	R54C	HTN	IS, HS	+/+	Lt. pons	Lt. BG	23	B,C,D,E	3	a2,b2,c
22	49/F	160C>T	11	R54C		migraine	+/+			0		3	a2,b2,c
23	70/F	1624T>C	11	C542R*		IS, migraine	+/+	Rt. thalamus		11	B,C,D,E	3	a2,b2,c
24	50/F	1624T>C	11	C542R*		–	+/−			N/A	N/A	N/A	N/A
25	60/F	1630C>T	11	R544C		–	+/+			0		2	c
26	55/M	1630C>T	11	R544C	HTN	migraine	−/−	N/A	N/A	N/A	N/A	N/A	c
27	43/F	1630C>T	11	R544C	DL	– (MMD)	+/+			1	B	1	c
28	62/F	1630C>T	11	R544C		IS	+/+	Lt.thalamus		10	B,E	2	a1,b1,c
29	64/M	1759C>T	11	R587C		HS	+/+		Lt. thalamus	2	E	2	c
30	55/M	3182G>A	20	C1061Y	HTN, DL	IS	+/+	Rt. MCA		41	A,B,C,D,E	3	a2,b2,c
31	53/M	3226C>T	20	R1075C*		–	+/+			6	B,E	3	a1,b2,c
32	68/M	3523C>T	22	R1175W*	HTN, DL, SM	–	+/+			0		3	a2,b2,c
33	46/F	3523C>T	22	R1175W*	HTN, DM, DL	IS, HS	+/+	Lt. MCA	Lt. CR	0		1	–
34	35/F	3523C>T	22	R1175W*		HS, (MMD)	+/+		Lt. BG	0		0	–

Location of MBs: A, Basal ganglia; B, thalamus; C, cerebellum; D, brainstem; E, subcortex. White matter lesions: a1, anterior temporal lobe, unilateral; a2, anterior temporal lobe, bilateral; b1, external capsule, unilateral; b2, external capsule, bilateral; c, periventricular white matters. * Novel variants of unknown significance. † Three patients (number 17–19) are one family. Patient No. 18 was a daughter of patient No.17 and patient No. 19 was a sister of patient No.17. Patient No. 26 did not have brain MRI images, and MRI results from other hospitals were available only on chart records. Abbreviations: DL, dyslipidemia; HTN, hypertension; DM, diabetes mellitus; SM, smoking; IS, ischemic stroke; HS, hemorrhagic stroke; MMD, moyamoya disease; Rt, right; Lt, left; MCA, middle cerebral artery; CR, corona radiata; BG, basal ganglia; N/A, not applicable; MBs, microbleeds; MRI, magnetic resonance image.

**Table 2 medicina-55-00521-t002:** Comparison of the different spectrum of NOTCH3, MRI features, and clinical presentations of cerebral autosomal dominant arteriopathy with subcortical infarcts and leukoencephalopathy (CADASIL).

Study	Race	Number of Patients	Age of Onset (± SD)	Mutation of NOTCH3	MRI Features	Clinical Presentations				
						Stroke	Cognitive Impairment	Psychiatric Syndrome	Headaches	Seizures
This study	Korean	34	52.5 ± 9.5	Exon 2–6: 41.2% (R75P/Q*: 17.6%) Exon 11: 26.5%	AT: 50% EC: 55.9%	IS: 44.1% ICH: 17.6%	Not specified	0	2.5%	0
Markus et al., 2002 [2]	British	48	35.9 ± 14.6	Exon 2–6: 93.8%	AT:89% EC: 93%	IS: 29.2% ICH: not specified	2.1%	8.3%	54.2%	4.2%
Desmond et al., 1999 [10]	American	105	36.7 ± 12.9	Not specified	Not specified	IS: 42.9%	5.7%	8.6%	40%	2.9%
Dotti et al., 2005 [12]	Italian	28	Not specified	Exon 2–6: 46.42% Exon 11: 21.4%	Not specified	Not specified	Not specified	Not specified	Not specified	Not specified
Lee et al., 2009 [5]	Han Chinese in Taiwan	21	48.6 ± 13.8	Exon 2–6: 28.6% Exon 11 (R544C): 47.6%	AT: 42% EC: 95.2%	IS: 52.4% ICH: 23.8%	4.8%	9.5%	4.8%	4.8%
Liu et al., 2015 [4]	Chinese mainland	62	39.7 ± 8.03	Exon 4: 59.6% Exon 3: 22.8% Exon 11: 3.5%	AT: 63.5%* EC: 69.2%*	IS or TIA: 75.8%	11.3%	3.2%	8.1%	Not specified
Choi et al., 2006 [7]	Korean	20	57.2 ± 10.2	Exon 2–6 (R75P): 10% Exon 11: 85% (R544C: 75%)	AT: 20% EC: 90%	IS: 55% ICH: 25%	15%	0	10%	0
Choi et al., 2013 [17]	Korean	73	62.7 ± 11.1	Exon 2–6 (R75P): 2.7% Exon 11: 95.9% (R544C: 90.3%)	Not specified	IS: 42.5% ICH: 12.3%	Not specified	Not specified	Not specified	Not specified
Kim et al., 2006 [6]	Korean	27	47.7	Exon 2–6: 77.8% (R75P: 55.6%) Exon 11: 22.2%	AT: 22.2% EC: 51.9%	IS: 40.7% ICH: 33.3%	18.5%	Depression 14.8%	3.7%	0
Ueda et al., 2015 [8]	Japanese	70	Patients with R75P: 53.6 ± 6.9 Patients with other mutations: 44.2 ± 12	Exon 3: 21% Exon 4: 69%	AT: 70.6%† EC: 76.5%†	IS or TIA: 69%	31%	20%	33%	Not specified

Abbreviations: SD, standard deviation; AT, anterior temporal; EC, external capsular; IS, ischemic; ICH, intracerebral hemorrhage. * Frequency was calculated among 52 patients with CADASIL Scale scores. † Frequency was calculated among 51 CADASIL patients with available MRI data.

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
