# Peer review of "Novel Characteristics of Race-Specific Genetic Functions in Korean CADASIL"

_medicina, 2019, doi:10.3390/medicina55090521_

Round 1
Reviewer 1 Report
The manuscript entitled " Novel Characteristics of race specific Genetic Functions in Korean CADASIL" is a well written and well documented genetic study, which concluded that the associations between HS and common genetic locations account for the increased development of intracranial hemorrhages in Korean poeple in comparison with Caucasians. The abstract is imformative, the methods are adequately described and the conclusions are supported by the results.The bibliography is relevent and well discussed. In the future a further clarification of the variable functional roles of NOTCH3 mutations would enlarge the horizons of the study.
Author Response
Response: Thanks for your valuable review.
As reviewers have said, I hope this study will help to clarify the functional role of NOTCH in the future.
Reviewer 2 Report
The paper “Novel Characteristics of Race-Specific Genetic Functions in Korean CADASIL” concerns with the analysis of a cohort of Korean patients with cerebral autosomal dominant arteriopathy with subcortical infarcts and leukoencephalopathy (CADASIL). Novel variants have been identified and race-specific features have been described.
However the objective of a previous paper by Kim et al (doi:10.1212/01.wnl.0000216259.99811.50) (ref 7 in the submitted paper) was: “To elucidate the phenotype, genotype, and MRI findings of Korean patients with cerebral autosomal dominant arteriopathy with subcortical infarcts and leukoencephalopathy (CADASIL) and mutation carriers”
Moreover, those authors stated: “Although Korean cerebral autosomal dominant arteriopathy with subcortical infarcts and leukoencephalopathy (CADASIL) mutation carriers show similar clinical and MRI findings, these abnormalities appear less frequently than in other populations”.
Overall, the present paper lacks of novelty, probably its only merit is to enlarge the number of studied cases.
In this frame, I think that this paper could be published.
I would suggest to rephrase “Therefore, our aim of this study was to evaluate correlations between genetic and clinical/radiological findings in Korean patients” in order to take into account previous reports concerning Korean patients.
Moreover, I suggest rephrasing the sentence “While all patients with mutations in exon 4 had AT lesions, approximately less than a third of 94 patients with mutations in R75P/R75Q VUS (16.7%), Y465C (0%), R544C (25%) and R1175W VUS (33.3%) had AT lesions.” It is not clear to me its meaning.
Please give more details in the legend to figure 1. What is the meaning of the mutations written in red?
Author Response
Reviewer #2:
The paper “Novel Characteristics of Race-Specific Genetic Functions in Korean CADASIL” concerns with the analysis of a cohort of Korean patients with cerebral autosomal dominant arteriopathy with subcortical infarcts and leukoencephalopathy (CADASIL). Novel variants have been identified and race-specific features have been described.
However the objective of a previous paper by Kim et al (doi:10.1212/01.wnl.0000216259.99811.50) (ref 7 in the submitted paper) was: “To elucidate the phenotype, genotype, and MRI findings of Korean patients with cerebral autosomal dominant arteriopathy with subcortical infarcts and leukoencephalopathy (CADASIL) and mutation carriers”
Moreover, those authors stated: “Although Korean cerebral autosomal dominant arteriopathy with subcortical infarcts and leukoencephalopathy (CADASIL) mutation carriers show similar clinical and MRI findings, these abnormalities appear less frequently than in other populations”.
Overall, the present paper lacks of novelty, probably its only merit is to enlarge the number of studied cases. In this frame, I think that this paper could be published.
I would suggest to rephrase “Therefore, our aim of this study was to evaluate correlations between genetic and clinical/radiological findings in Korean patients” in order to take into account previous reports concerning Korean patients
Response: Thanks for your valuable feedback.
As the reviewer said, the previous paper has already reported the CADASIL features in Korean. However, several findings on specific genetic regions associated with hemorrhagic stroke or reports on variants of unknown significance (VUS), are novel in this paper.
Following the reviewer's advice, we changed the sentences as follows;
In abstract, Background and objectives,
“Therefore, we sought to investigate the correlations between genetic & clinical/radiological findings in Korean CADASIL patients including some variants of unknown significance.”
Moreover, I suggest rephrasing the sentence “While all patients with mutations in exon 4 had AT lesions, approximately less than a third of patients with mutations in R75P/R75Q VUS (16.7%), Y465C (0%), R544C (25%) and R1175W VUS (33.3%) had AT lesions.” It is not clear to me its meaning.
Response: Thanks for your valuable feedback.
We changed the sentences as follows.
In Result, Paragraph 4, Line 1,
While all patients with mutations in exon 4 had AT lesions, approximately less than a third of patients with mutations in some mutational hotspots had AT lesions; R75P/R75Q VUS (16.7%), Y465C (0%), R544C (25%) and R1175W VUS (33.3%).
Please give more details in the legend to figure 1. What is the meaning of the mutations written in red?
Response: Thanks for your detailed comments.
The red text is the most common genetic region.
We added the sentence in Figure 1 legend as follows.
Figure 1. NOTCH3 gene mutations. (a) NOTCH3 gene located on chromosome 19p13.12; (b) NOTCH3 gene mutations are located within the extracellular domain that encodes the epidermal growth factor like repeats, 3 Lin-NOTCH repeats, and 1 transmembrane region; (c) NOTCH3 gene mutations or variants of unknown significances in our study. Red bold text is the most common genetic region in this study. *Novel variants of unknown significance
Reviewer 3 Report
This is a novel study examining the genetic mechanisms and the clinical/radiological findings in Korean patients with CADASIL. It provides important information to the understanding of the genetic background of CADASIL, especially in the Asian population. I have a few questions:
In Introduction, the authors mentioned that the NOTCH 3 mutations were found in exon 11 in Taiwan and in Korea. Is there any report showing a similar genetic-radiologic correlation in the Taiwanese CADASIL population? This could be included as part of the discussion. In Methods “We screened 198 consecutive patients with a suspected diagnosis of CADASIL” Could the authors further explain how the patients were screened- who made the diagnosis of suspected CADASIL, was it based on clinical or radiological findings? In addition, please further explain how subjects were included/excluded. Page 2, line 62 “MRI was performed at 1.5-T or 3.0-T in almost patients.”. Please rephrase this sentence. In Results: The authors listed the symptoms that subjects were having. How many of the subjects had cognitive decline? HS was found to be associated with mutations in 4 exons. How about IS, is it associated with mutations in the same exons too? Table 1 is an excellent summary. It looks like two patients did not have MRI. Could the authors explain the reasons? The authors mentioned AT and EC hyperintensities are thought to be markers for Caucasian CADASIL patients but not Asians. Is there any unique imaging features for Asian CADASIL patients that the authors would propose from this study and literature review? In Discussion, the authors stated the majority of mutations were associated with less prevalent AT or EC hyperintensities. In the results, such as figure 3, AT involvement was discussed. Could the authors provide more information on the results for the association between the mutations and EC hyperintensities?
Author Response
Reviewer #3:
This is a novel study examining the genetic mechanisms and the clinical/radiological findings in Korean patients with CADASIL. It provides important information to the understanding of the genetic background of CADASIL, especially in the Asian population. I have a few questions:
In Introduction, the authors mentioned that the NOTCH 3 mutations were found in exon 11 in Taiwan and in Korea. Is there any report showing a similar genetic-radiologic correlation in the Taiwanese CADASIL population? This could be included as part of the discussion.
Response: Thanks for your detailed comments.
We added the sentences as follows.
In Discussion section, Paragraph 2, Line 8,
There are some Taiwan studies that analyze stroke etiology and radiological findings focused on R544C mutations. According to a prospective, multicenter study in Taiwan, small vessel occlusion was significantly associated with R544C mutation. Another Taiwan study demonstrated that of 67 patients with R544C mutations, patients with ICH-onset were more likely to have whitematter hyperintensities and higher total CMBs.
In Methods “We screened 198 consecutive patients with a suspected diagnosis of CADASIL” Could the authors further explain how the patients were screened- who made the diagnosis of suspected CADASIL, was it based on clinical or radiological findings? In addition, please further explain how subjects were included/excluded.
Response: Thanks for your detailed comments.
We added the sentences as follows.
In Materials and Methods section, Paragraph 1, Line 1,
We screened 198 consecutive patients with a suspected diagnosis of CADASIL between 2005 and 2015 at Seoul National University Hospital (SNUH). Genetic work-ups were performed based on the physicians’ discretion considering the previous medical history, neurological examination, and MRI findings. The inclusion criteria were severe whitematter hyperintensities on the brain MRI and at least one of the following: recurrent strokes at relatively young age, cognitive impairment, psychiatric symptoms, migraine, seizure and family history of strokes. Patients with definite etiologies for index stroke lesions were excluded. Page 2, line 62 “MRI was performed at 1.5-T or 3.0-T in almost patients.”. Please rephrase this sentence.
Response: Thanks for your detailed comments.
We changed the sentences as follows.
In Materials and Methods, Paragraph 3,
1.5 Tesla or 3.0 Tesla MRI was performed in almost patients except 2 subjects. One was a daughter of a symptomatic patient. Since she did not have any clinical symptoms, she did not want to take brain image. Although another subject did not have MRI image, MRI results from other hospitals were available only on chart records.
In Results: The authors listed the symptoms that subjects were having. How many of the subjects had cognitive decline?
Response: Thanks for your valuable comments.
Unfortunately, since most patients were examined during the acute ischemic stroke, cognitive evaluations were not sufficient. We added this limitation in discussion section as follows.
In discussion section, paragraph 6, Line 5,
Third, this was a retrospective study. Since most patients were examined during the acute stroke period, cognitive evaluations were not sufficient.
HS was found to be associated with mutations in 4 exons. How about IS, is it associated with mutations in the same exons too?
Response: Thanks for your valuable comments.
In fact, in this study, none of the patients with NOTCH3 mutations in exon 4 had HS. We cannot find of correlations between phenotypes and genotypes. However, in previous articles involving a large number of patients with exon 4 genomic abnormalities, ischemic stroke was more common.
References>
Ueda, A.; Ueda, M.; Nagatoshi, A.; Hirano, T.; Ito, T.; Arai, N.; Uyama, E.; Mori, K.; Nakamura, M.; Shinriki, S., et al. Genotypic and phenotypic spectrum of CADASIL in Japan: the experience at a referral center in Kumamoto University from 1997 to 2014. Journal of neurology 2015, 262, 1828-1836,
Liu, X.; Zuo, Y.; Sun, W.; Zhang, W.; Lv, H.; Huang, Y.; Xiao, J.; Yuan, Y.; Wang, Z. The genetic spectrum and the evaluation of CADASIL screening scale in Chinese patients with NOTCH3 mutations. Journal of the neurological sciences 2015, 354, 63-69, doi:10.1016/j.jns.2015.04.047.
We presented these findings in Table 2.
Table 1 is an excellent summary. It looks like two patients did not have MRI. Could the authors explain the reasons?
Response: Thanks for your valuable comments.
As I mentioned in the footnote of Table 1,
“Three patients (number 17-19) are one family. Patient No. 18 was a daughter of patient No.17 and patient No. 19 was a sister of patient No.17.”
Since Patient No. 18 is still young and does not have any clinical symptoms, she doesn’t want to perform brain image.
Patient No. 26 did not have MRI images, and MRI results from other hospitals were available only on chart records.
We added these sentences as follows.
In Table 1, Footnote,
“Patient No. 26 did not have MRI images, and MRI results from other hospitals were available only on chart records.”
In Method section, Paragraph 3, Line 1,
1.5 Tesla or 3.0 Tesla MRI was performed in almost patients except 2 subjects. One was a daughter of a symptomatic patient. Since she did not have any clinical symptoms, she did not want to take brain image. Although another subject did not have MRI image, MRI results from other hospitals were available only on chart records.
The authors mentioned AT and EC hyperintensities are thought to be markers for Caucasian CADASIL patients but not Asians. Is there any unique imaging features for Asian CADASIL patients that the authors would propose from this study and literature review?
Response: Thanks for your valuable comments.
Unfortunately, to date, no specific MRI markers have been identified in Asian CADASIL patients. However, in some studies including Asian population, it is suggested that microbleeds and hemorrhagic stroke are more frequent in Asian subjects.
References>
Kim, Y.; Choi, E.J.; Choi, C.G.; Kim, G.; Choi, J.H.; Yoo, H.W.; Kim, J.S. Characteristics of CADASIL in Korea: a novel cysteine-sparing Notch3 mutation. Neurology 2006, 66, 1511-1516, doi:10.1212/01.wnl.0000216259.99811.50.
Choi, J.C.; Kang, S.Y.; Kang, J.H.; Park, J.K. Intracerebral hemorrhages in CADASIL. Neurology 2006, 67, 2042-2044, doi:10.1212/01.wnl.0000246601.70918.06.
We mentioned these findings as follows.
In Discussion, Paragraph 3, Line 9,
Compared to those patients, none of the patients with NOTCH3 mutations in exon 4 had HS. These findings may account for the increased development of ICH in Korean individuals rather than Caucasians [6].
In Discussion, the authors stated the majority of mutations were associated with less prevalent AT or EC hyperintensities. In the results, such as figure 3, AT involvement was discussed. Could the authors provide more information on the results for the association between the mutations and EC hyperintensities?
Response: Thanks for your valuable comments.
To clarify the proportion, we made the following figure.
However, because the patterns are similar with AT hyperintensities, we did not include this figure on the manuscript. Instead, we added the sentences as follows.
In Result section, Paragraph 4, Line 5,
Results with EC hyperinetensities also showed similar distribution.